# Motivational Profiles in Physical Education: Differences at the Psychosocial, Gender, Age and Extracurricular Sports Practice Levels

**DOI:** 10.3390/children10010112

**Published:** 2023-01-05

**Authors:** Diego Andrés Heredia-León, Alfonso Valero-Valenzuela, Alberto Gómez-Mármol, David Manzano-Sánchez

**Affiliations:** 1Faculty of Sport Sciences, University of Murcia, 30100 Murcia, Spain; 2Academic Unit of Education, Catholic University of Cuenca, Cuenca 010105, Ecuador; 3Faculty of Education, University of Murcia, 30100 Murcia, Spain

**Keywords:** theory of self-determination, motivation, basic psychological needs, cluster analysis, students

## Abstract

The objective of this study was to analyze the motivational profiles of Physical Education (PE) students and assess differences based on the perception of the support of autonomy, the intention to be physically active, satisfaction with classes, gender, age, and extracurricular sports activities. A cross-sectional descriptive study was carried out with a sample of 2621 students, aged 8 to 18 years (*M* = 14.16, *SD* = 2.28). An analysis of profiles was implemented, taking into account the motivation and the index of psychological mediators. The result of the cluster analysis gave a solution to four motivational profiles: high quality (*n* = 1094), low quantity (*n* = 292), low quality (*n* = 555), and high quantity (*n* = 680). Students grouped in the high quantity and quality profile presented higher levels of autonomy support, physical activity intention, enjoyment, and lower levels of boredom. Male participants, younger students, and those who participated in extracurricular activity were associated with self-determined profiles. In conclusion, the importance of promoting the satisfaction of basic psychological needs and autonomous motivation in PE classes is highlighted, in order to achieve higher values in terms of autonomy support, the intention to be physically active, enjoyment, and to reduce boredom levels in students.

## 1. Introduction

Motivation has been extensively investigated in the field of Physical Education (PE), as sport is one of the determining and influencing factors in the behavior of human beings [1]. In the last two decades, many studies have been based on the conceptual framework of the Theory of Self-Determination (SDT) [2,3,4], which suggests that the interaction between the individual and the surrounding context is key for motivation, behavior and well-being [2,3]. SDT considers the level at which behaviors are self-determined or not, that is, to what extent people perform their exercises voluntarily [5]. According to Deci and Ryan [2], this theory establishes different types of motivation along a continuum, depending on the level of self-determination.

Thus, intrinsic motivation, extrinsic motivation and amotivation can be found, to a greater or lesser extent, with respect to self-determination. To establish more specific dimensions, Deci and Ryan [6] proposed that the regulation of behavior could be differentiated into three groups: autonomous motivation, which is composed of intrinsic motivation; integrated and identified regulations; controlled motivation, which is assembled by introjected and external regulations; and amotivation.

Motivation has been studied through motivational profiles employing a cluster analysis from different approaches, such as in the field of PE [7,8,9], sports [5,10,11] and university students [12,13,14], among others.

SDT encompasses, in turn, the Basic Psychological Needs (BPN) theory, which highlights the presence of three psychological needs: competence (focused on mastery of skills), autonomy (oriented to the development of good actions) and relationship (aimed at connecting and giving affection to others) which, when developed, promote progress in motivation [15]. Thus, in this type of environment, these needs will be satisfied, while, on the contrary, in controlling contexts, needs frustration appears [3,16]. Several studies found differences based on motivation profiles in this variable [11,17], showing that students who developed BPN were the ones with more self-determined profiles.

Studies such as Almagro et al. [5] identify the existence of different patterns in students’ behavior which produce relevant information that should be provided for their teachers. In addition, there are also other studies where not only the different types of motivation were included in the construction of profiles, but BPN as well [18,19], with the aim of identifying the changes in support for autonomy, the intention to be physically active and sports practice satisfaction.

Regarding the importance of autonomy support, Williams et al. [20] explain how the degree to which teachers recognize the ability of students and encourage their autonomous and active participation in learning can improve students’ intrinsic motivation [21,22]. Further studies have found differences in this dimension based on motivational profiles [18,23,24].

Concerning PE lessons, it is imperative to acknowledge the importance of the intention to be physically active, generating adequate environments that foster motivation not only during the lessons themselves, but also in other contexts and at later ages [25,26].

With reference to enjoyment and boredom, Baena-Extremera et al. [27] state that these variables can be determinantal in one’s interest in doing sports, with the consequent importance that this has in the acquisition and adherence to physical exercise habits. Enjoyment plays a key role in promoting the internal motivation of students, characterized by the experience of happiness and enthusiasm in a situation that is entertaining [28].

The present work focuses on analyzing motivational profiles regarding gender, age and out-of-school sports practice. In the case of gender, previous studies have shown that men show higher values in self-determined motivation compared to women [29,30,31]. However, some motivational-undetermined profiles were associated [9,32] leaving no efficiently conclusive studies. Concerning age, researchers reported there was a greater number of younger students associated with profiles of low quantity and quality compared to older students who were associated with high-quality profiles [32]. Contrary to, this, in the study by Manzano-Sánchez et al. [9] no differences were found based on age in the quality ranges of the profiles. In out-of-school sports activity, Moreno et al. [19] mention that such students are aligned with the most self-determined profiles; this same result was found by Granero-Gallegos et al. [30], but in PE students. Research based on gender, age and extracurricular practice is still in development, so there are many keys that this study could solve.

Following the analysis of the literature, the importance of motivation in Spanish participants is glimpsed. To know the different motivational profiles and their differences according to sociodemographic variables and those related to physical activity can help us to find out if the profiles found in previous research can be found in Ecuadorian participants, in addition to knowing the changes according to gender, age and extracurricular practice.

The objective of this research was, firstly, to analyze the motivation profiles that could exist in a sample of PE students, including the different types of motivation, as well as introduce the satisfaction of BPN. Secondly, the research checks, through these motivational profiles, the potential differences in support for autonomy, intention to be physically active and sports practice satisfaction. Lastly, the research set out to analyze the sample’s motivational profiles based on gender, age and out-of-school practice.

It is hypothesized that there are different motivation profiles at the psychosocial level affecting gender, age and extracurricular sports practice, where the profiles that have a more self-determined motivation will be those that have higher values in support of autonomy, and an intention to be physically active and fun while the profiles that have a non-self-determined motivation will be the ones that have higher values in boredom. Students with a high-quality profile of self-determined motivation are going to be the older ones, of the male gender, and the ones that do practice extracurricular activities.

## 2. Materials and Methods

### 2.1. Design

This research corresponds to a cross-sectional quantitative design [33], whose design and development was favored by the Research Ethics Commission of the University of Murcia, code 3023/2020.

### 2.2. Participants

The sample consisted of 2795 students from the fifth year of elementary school to the third year of high school from five public educational units located in urban areas of four provinces of Ecuador, selected at convenience. After omitting the questionnaires that were not completed in their entirety and implementing the statistical procedures to detect atypical cases and missing values, the final sample was composed of 2621 male students (*n* = 1303) making up 49.70% and female (*n* = 1318) with 50.30%, aged between 8 and 18 years old (*M* = 14.16, *SD* = 2.28), of which 620 (23.70%) students were in an age range of 8 to 12 years old, 1227 (46.80%) between 13 to 15 years old and 774 (29.50%) between 16 to 18.

Of the total number of participants, 1453 (55.40%) carried out some type of extracurricular sports activity, trained between 1 and 7 days a week, (*M* = 3.82, *SD* = 1.60), the duration of the training ranged from 1 to 5 h (*M* = 1.89, *SD* = 0.96), divided between 1 to 3 sessions per day (*M* = 1.47, *SD* = 0.65), while 1168 (44.60%) did not participate in any type of physical or sport activity outside regular class hours.

### 2.3. Instruments

#### 2.3.1. Autonomy Support

The Spanish version of The Learning Climate Questionnaire (LCQ; [34]), validated in Spanish by Núñez et al. [35] was used; the long version was applied, in which are found 15 items preceded by the phrase “My physical education teacher…,” to appraise the single dimension of autonomy support (e.g., “answers my questions carefully and in detail”). Answers were scored on a Likert-type scale ranging from 1 (completely disagree) to 7 (completely agree). The internal consistency in the present study revealed a value of α = 0.85, Ω = 0.87.

#### 2.3.2. Basic Psychological Needs

The Psychological Need Satisfaction in Exercise Scale (PNSE) by Wilson et al. [36] was applied, validated in the Spanish context by Moreno-Murcia et al. [37]. The PNSE uses eighteen items, six to assess each of the needs: competence (e.g., “I feel confident in my ability to perform exercises that challenge me”), autonomy (e.g., “I feel free to choose which exercises I will take part”) and relationship with others (e.g., “I feel close to my exercise companions who appreciate how difficult exercise can be”). The previous sentence was “In my physical education classes…” and the answers were gathered on a Likert-type scale, whose score ranges between 1 (False) and 6 (True). Internal consistency in the present study revealed a value of α = 0.89, Ω = 0.89 for competence, α = 0.87, Ω = 0.87 for autonomy, and α = 0.86, Ω = 0.87 for relationship with others. The Psychological Mediators Index (IMP) was developed by calculating the mean of the three factors in a single dimension, the internal consistency was α = 0.79, Ω = 0.80.

#### 2.3.3. Motivation

The questionnaire called Sport Conduct Regulation Questionnaire (BRSQ) by Lonsdale et al. [38], validated in Spanish by Moreno-Murcia et al., was used [37]. It comprises thirty-six items grouped into nine factors of four items each that measure general intrinsic motivation (e.g., “Because I find it pleasurable”), intrinsic motivation concerning knowledge (e.g., “I enjoy learning something new about my sport”), stimulation in intrinsic motivation (e.g., “Because of the pleasure I experience when I feel completely absorbed in my sport”), intrinsic motivation regarding achievement (e.g., “Because I get a sense of accomplishment when I strive to achieve my goals”), integrated regulation (e.g., “Because what I do in sport is an expression of who I am”), identified regulation (e.g., “Because it is a good way to learn things which could be useful to me in my life”), introjected regulation (e.g., “Because I would feel guilty if I quit”), external regulation (e.g., “Because people push me to play”) and amotivation (e.g., “But I question why I am putting myself through this”). The introductory phrase used was: “I participate in my physical education classes…”. A seven-point Likert-type scale was used, ranging from 1 (completely false) to 7 (completely true). Following the contributions made by different authors (Deci and Ryan) [6], and Vansteenkiste et al. [39], the different dimensions were grouped into three factors: autonomous motivation (intrinsic and identified), controlled motivation (introjected and external) and amotivation. The reliability values of the variables in the present study was α = 0.86, Ω = 0.86 for general motivation, α = 0.90, Ω = 0.90 for intrinsic knowledge motivation, α = 0.90, Ω = 0.90 for stimulation in intrinsic motivation, α = 0.78, Ω = 0.78 for achievement in intrinsic motivation, α = 0.78, Ω = 0.78 for integrated regulation, α = 0.87, Ω = 0.87 for identified regulation, α = 0.93, Ω = 0.93 for introjected regulation, α = 0.93, Ω = 0.93 for external regulation, α = 0.94, Ω = 0.94 for amotivation, for the calculated factors it was α = 0.97, Ω = 0.97 for autonomous motivation and α = 0.94, Ω = 0.94 for controlling motivation.

#### 2.3.4. Future Intention to Be Physically Active

The questionnaire used is called “Intention to be physically active” (IPA) by Hein et al. [40] and validated in the Spanish context by Moreno, et al. [41]. This questionnaire consists of 5 items. (e.g., “Outside PE lessons I like to do sport”). The previous sentence used was: “Regarding your intention to practice some physical-sports activity…”. The answers were closed with a Likert-type scale whose score ranged from totally disagree (1) to totally agree (5). The reliability value in the present study was α = 0.80, Ω = 0.80.

#### 2.3.5. Sport Satisfaction

The Spanish version of Balaguer et al. [42] of the Sport Satisfaction Instrument (SSI) by Duda and Nicholls [43] was used, adapted to EF (SSI-EF) by Baena-Extremera et al. [27]. The SSI-EF consists of 8 items to measure intrinsic satisfaction in a sports activity through two subscales that measure satisfaction/enjoyment (5 items) and boredom (3 items) in sports activity. The scale was preceded by the phrase “Indicate your degree of disagreement or agreement with the following statements, referring to your Physical Education classes…”. An example of a satisfaction/enjoyment item was (e.g., “I usually get involved in the Physical Education classes”) and a boredom item was (e.g., “In Physical Education classes, I am usually bored”). Responses were collected on a 5-point Likert-type scale ranging from strongly disagree (1) to strongly agree (5). The internal consistency analysis in the present study for enjoyment was α = 0.87, Ω = 0.87 and for boredom α = 0.65, Ω = 0.66.

#### 2.3.6. Gender, Age and Extracurricular Sports Activity

The following variables were used to corroborate their interaction with the found profiles: Gender, categorized as Man or Woman; Age, categorized into three groups (8 to 12 years, 13 to 15 years and 16 to 18 years); and the level of sports practice outside school hours was assessed through the question: “Do you do any physical-sports activity outside of class hours?”, a question which the students had to answer “yes” or “no”. If they answered yes, then they would also answer, how many times a week, as well as how many hours a week, and how many sessions a day.

### 2.4. Procedure

The management of the teaching centres and the PE teachers were contacted to request their authorization, informing them of the objectives and intentions of the research project. Subsequently, the parents/guardians of the students were informed and asked to sign an informed consent. The anonymity of the information was communicated to the students, in exchange for which they were asked for maximum sincerity. The questionnaires were applied through the Google Docs extension, the responses lasted 15–25 min and were completed in classrooms equipped with internet access and with the presence of the researcher, who expressed the possibility of clarifying any doubts during the process.

### 2.5. Data Analysis

As a first point, to examine the reliability of each variable analyzed, Cronbach’s alpha coefficient was estimated, and McDonald’s omega was calculated following the recommendations of Hayes and Coutts [44], who consider its reliability analysis to be greater, since it works with factor loadings which make the calculations more stable and presents a higher confidence level of reliability, being acceptable with values greater than 0.70 [45]. Next, the Mahalanobis distance was used in order to detect and eliminate those atypical subjects or those who did not follow a logical pattern in the set of variables. In addition, the values of skewness and kurtosis were analyzed, less than two in asymmetry and seven in kurtosis [46] being considered appropriate. Once those subjects who did not follow a logical pattern in the studied variables were eliminated, a final sample of 2621 subjects was obtained and the reliability analysis of the scales was carried out again, which remained above 0.70, with only boredom at 0.65. Subsequently an analysis of bivariate correlations was carried out among all the subscales under study.

Subsequently, a cluster analysis was performed, using autonomous motivation, controlled motivation, amotivation, and NPBs united in a single factor (autonomy, competence, and social relationships) to establish the motivational profiles. Following Hair et al. [47] a two-step hierarchical and non-hierarchical cluster analysis was performed using Ward’s model and Z-scores. The dendrogram suggested four profiles, and the hypothesis of four groupings was confirmed with the consistent results for this solution from the k-means method. In addition, a double-split cross-validation approach was performed to inspect for stability by randomly splitting the sample into halves and applying the procedure again to each subsample.

Next, the multivariate analysis (MANOVA) was carried out to check whether there were statistically significant differences in each of the variables under investigation. In the cases in which significant differences were found, a contrast test of post hoc comparisons was carried out using the Bonferroni correction to determine the clusters between which there would be differences. In addition, the effect size was calculated in terms of partial eta squared (η^2^), considering a small effect size with values between 0.01 and 0.05, a medium effect between 0.06 and 0.13, and a large effect with a value 0.14 or above [48]. The Box test was used prior to this process to analyze the homogeneity of covariances [49].

Finally, a Pearson chi-square test was carried out, together with a residual analysis, to examine the potential differences between the profiles found based on gender, age and extracurricular sports activity. Statistical analyses were performed using the IBM SSPS 24.0 package. This established a level of significance of *p* < 0.05.

## 3. Results

### 3.1. Descriptive Analysis, Reliability and Correlations

Table 1 shows the descriptive analyses of the different variables under study, the mean, standard deviation, asymmetry values, kurtosis, reliability and bivariate correlations. All the variables had a reliability value > 0.70, with the exception of the boredom dimension whose value was 0.65, a level considered acceptable by Sturmey et al. [50].

In terms of correlations, autonomy support was positively correlated with the three basic psychological needs: with intrinsic motivation (general, knowledge stimulation and achievement), with integrated and identified regulations, and with the intention to be physically active, but it was not correlated with introjected and external regulation and was negatively correlated with amotivation and boredom. There was, however, a positive correlation between boredom, introjected and external regulations, and amotivation.

### 3.2. Cluster Analysis and Obtaining Motivational Profiles

Firstly, the conglomerates’ hierarchical analysis was carried out with the Ward method, and as a result, a dendrogram and the agglomeration coefficients were obtained reflecting the existence of three possible grouping solutions, which were: (1) two profiles, low quality and high quality; (2) three profiles, high quantity, low quality, and high quality; and (3) four profiles: low quantity, high quantity, low quality, and high quality. The decision on the optimal number of profiles is subjective and determined, on the one hand, by the distance matrix, and on the other, by the theory that supports the realization of these profiles.

In such manner, based on the results of previous studies Manzano-Sánchez et al. [9], Sánchez-Oliva et al. [32] and considering the main theoretical contributions made in this regard by Vansteenkiste et al. [51] and Vansteenkiste et al. [39], the structure made up of four profiles was chosen as the most convenient solution. Nonetheless, due to the exploratory nature of the hierarchical analysis, the results obtained were confirmed through another type of non-hierarchical technique. For this, a K-means cluster analysis was carried out which informs us of the degree of similarity between the number of clusters found in the hierarchical and non-hierarchical analyses. The results of the two analyses were very consistent, supporting the decision to employ four clusters.

As shown in Figure 1 and Table 2, the first of these profiles is named as high quality and groups 1094 participants (41.74%). This conglomerate is characterized by having high values in autonomous motivation and NPB, but very low values in amotivation and controlling motivation. The second profile is made up of 292 participants (11.14%) and they belong to the low quantity profile. This profile stands out because the participants perceive slightly low levels in amotivation and controlling motivation, with very low values in autonomous motivations and BPN. The third profile is made up of 555 participants (21.18%) who would belong to the low-quality profile. This profile is characterized by students who perceive high values in amotivation and controlling motivation and low values in autonomous motivation and BPN. The fourth profile is called high quantity, and is made up of 680 students (25.94%).

These students stand out for perceiving high levels of motivation and controlling motivation, very high in autonomous motivation and BPN. Table 2 shows the differences between the variables that were part of the solution of the cluster. They had a multivariate effect (Box value = 1060.08, f = 35.21, *p* < 0.01), suggesting the use of the Pillai trace as a statistic test [49], showing a value of 1.55 (f = 699.09) *p* < 0.01.

### 3.3. Analysis of Differences among Clusters

The Box M and Levene tests, which measure the homogeneity of variance in the dependent variable in all the groups defined by the factors, were carried out, observing that only in the case of the autonomy support variable were the *p*-values greater than 0.05, so there was no homogeneity of variances for the rest of the variables, specifically the motivation variables (controlling and autonomous motivation, and amotivation), the basic psychological needs, enjoyment and boredom, and intention to be physically active. This leads us to indicate that not all the initial hypotheses were satisfied, and therefore, that the results are not completely conclusive.

The analysis of the multivariate test of variance (Manova) shows in Table 3 a statistically significant effect for the four motivational profiles, with Box M = 551.04 (F = 18.30, *p* < 0.01), and Pillai’s trace = 0.83, (F = 251.79, *p* < 0.01). All the motivational profiles described differ significantly from each other with respect to autonomy support, intention to be physically active, enjoyment and boredom. The results in Table 3 show the contrast of multiple comparisons using the Bonferroni correction.

The post hoc analysis reported that, in the case of autonomy support and enjoyment, differences were found among all the motivational profiles with the exception of the high-quality profile with high quantity. In the case of the intention to be physically active, differences were found among all four motivational profiles, obtaining higher values in the following order: high quantity, high quality, low quality, low quantity. For boredom there were differences among all the profiles with higher values in high quantity, followed by low quality, low quantity and high quality (Table 4).

### 3.4. Differences according to Gender, Age and Extracurricular Activity

To verify the existing differences in the distribution of the motivational profiles found in terms of gender, age and the practice of extracurricular physical activity, it was decided to carry out an analysis of differences using Pearson’s chi-square statistic. This test compares the observed and expected frequencies in each category to test whether all categories contain the same proportion of values or whether they contain a proportion of values specified by the user. Using the corrected standardized residuals gives us information on where these differences are found, since residuals equal to or greater than 1.96 are considered to be indicators that there are dependencies between these two categories and that the differences are therefore significant.

Table 5 reports the differences found in gender, where there were more boys (54.90%) than girls (45.10%) associated with high quantity profiles. In terms of age, there were more students aged 8–12 years associated with the high-quality profile (27.60%) as well as a lower prevalence of this group in the profile of low quantity (17.80%) and low quality (19.60%). Those over 16 years of age were associated with the low-quality profile (34.40%) and there was a lower preponderance of this group in the high-quality profile (27.20%). Finally, in extracurricular sports practice, there were higher frequencies in the profile of high quality and high quantity (60.40%), while the opposite occurred in the profile of low quantity (45.50) and low quality (44.70%).

## 4. Discussion

Based on the first of the proposed objectives, to analyze the motivational profiles that might exist in a sample of PE students, four types of motivational profiles were found. These results are in line with those found by Sánchez-Oliva et al. [32] in a sample made up of 1690 students and Manzano-Sánchez et al. [9] with 768 students, both groups with a mean age of 13, while Vergara-Morales et al. [14] found these same profiles among university students. However, results indicate no agreement with studies such as the one by Moreno et al. [52] who found three profiles, Haerens et al. [53] with five profiles, and Valle et al. [13], six profiles. This may be due to the different types of samples such as athletes, or the use of different measurement instruments of students to build these profiles.

The results obtained motivated a two-pronged classification, those of quality and quantity [39]. In the case of quality, the students who present a high-quality profile are those who, in PE classes, show greater autonomous motivation and BPN satisfaction and negative scores in controlling motivation and amotivation. On the other hand, with a low-quality profile, there are students with a controlling motivation and unmotivated towards PE, with a greater frustration of the need for autonomy, competence, and relationships with others.

In the case of quantity, the students who have negative values in amotivation, controlling motivation, autonomous motivation, and BPN are those who belonged to the low quantity profile, while those who have high values in the mentioned variables are those who belonged to the profile of high quantity. These results are in line with studies such as those by Manzano-Sánchez et al. [9] and Sánchez-Oliva et al. [32] where they consider the aspects of quality and quantity from four conglomerates and included the BPN when preparing the profiles.

Regarding the second of the objectives, we sought to verify the differences that exist in support for autonomy, intention to be physically active, enjoyment, and boredom based on the four types of established motivational profiles. The findings show that students with high-quality and high-quantity profiles had higher scores on autonomy support, intention to be physically active, and enjoyment, while low-quantity and low-quality profiles had lower scores. Boredom had lower values in the high-quality, low-quantity profile, and higher values in the low-quality, high-quantity profile. These results are in agreement with previous research, specifically, in the case of autonomy support with Fin et al. [18] and Valero-Valenzuela et al. [24], in the intention to be physically active with Franco et al. [8] and Huéscar et al. [17], and in enjoyment and boredom with Granero-Gallegos et al. [30] and Sánchez-Oliva, et al. [32], where students with autonomous and self-determined motivational profiles were the ones who developed greater autonomy support, intention to be physically active and enjoyment, while students who presented non-self-determined and unmotivated motivational profiles developed boredom.

The last of the objectives analyzed the motivational profiles according to gender, age, and extracurricular sports practice by the students, indicating that the boys presented higher values compared to the girls in the high quantity profile. The findings are shown to be in line with what was found by Sánchez-Oliva et al. [32] and Granero-Gallegos et al. [30] in a sample of 2002 Spanish students with an age range of 12 to 19 years. A point to note is that the girls have an excellent intensity in motivation, but their direction is less adequate, that is, they present high values in autonomous motivation, a very positive aspect, but also high values in controlled motivation and amotivation.

Concerning age, students between eight and twelve years old were associated with the high-quality profile and lower numbers in the profile of low quantity and low quality, while students between sixteen and eighteen years old were associated with the low-quality profile and lower presence in the high-quality profile. A study by Sánchez-Oliva et al. [32], on the contrary, found a greater number of younger students associated with profiles of low quantity and quality compared to older students who were associated with profiles of high quality. In the study by Manzano-Sánchez et al. [9] no differences were found in age.

Regarding extracurricular sports practice, it should be noted that most of the students who did some activity presented a high quantity and quality profile. Therefore, students who present a good quantity and quality of motivation towards PE are those who do more sports outside of class time, with results similar to those found in previous works, Moreno et al. [19] and Granero-Gallegos et al. [30].

The findings obtained in the present study may have a great practical application. In response to the first hypothesis, promoting a more self-determined motivation within physical education classes can work so that students perceive greater fun towards the practice of physical activity, with a greater intention to be physically active, not only in the present, but also in the future. This also translates into a greater perception of support for autonomy by students who obtained a motivational profile related to high quality, seeing the importance of teachers and their way of teaching to achieve adequate student training.

On the other hand, the results can help us to focus on intervention programs within physical education classes in order to promote extracurricular physical activity, where we find a more adaptive motivational profile in students. In the same way, focusing attention on early adolescence (thirteen–fifteen years old) seems to be appropriate since students between eight and twelve years old have a profile focused on high quality; this also presents the question: “What is happening in the transitions from primary education to secondary education”? Finally, it is noteworthy that men have a motivational profile oriented more towards quantity than quality, unlike women, emphasizing the importance of internal motivation, a necessary quality within the educational context of physical education.

As for the main limitations of this work, we must highlight the nature of the cross-sectional and descriptive study, one where cause-effect relationships cannot be established. On the other hand, the presence of the teacher may have influenced the responses of some students. Future research, especially of a longitudinal and experimental type [18,54,55], should be developed, as well as the practical implications of intervention programs in PE to establish teachers’ strategies in the development of NPB [54] and motivation [55], which could contribute to increasing support for autonomy, the intention of being physically active enjoyment and reducing boredom in PE classes. In addition, it would be interesting to analyze existing differences in these profiles in different samples, which could be university students or athletes, as well as considering the level of study and socioeconomic status.

## 5. Conclusions

It is concluded that there are different motivational profiles depending on the quality, quantity and type of motivation, as specified in four profiles. At a psychosocial level, support for autonomy, intention to be physically active and fun showed higher values in high quality and high quantity profiles, and negative values in low quality and low quantity profiles, while in the case of boredom the higher values were shown in low quality and high quantity profiles.

Regarding the socio-demographic variables that were studied, in the case of gender, men were associated with the high quantity profile, students between 8–12 years old were associated with the high quality profile and those between 16 and 18 years old with the low quality profile. Those who practiced extracurricular sports were located in the high quantity and high quality profiles compared to those who did not practice sports who were in the low quality and low quantity profiles.

Finally, an indication towards promoting BPN satisfaction and autonomous motivation in PE classes is essential, in order to promote support for autonomy, the intention to be physically active and enjoyment, as well as to reduce levels of boredom amongst students.

## Figures and Tables

**Figure 1 children-10-00112-f001:**
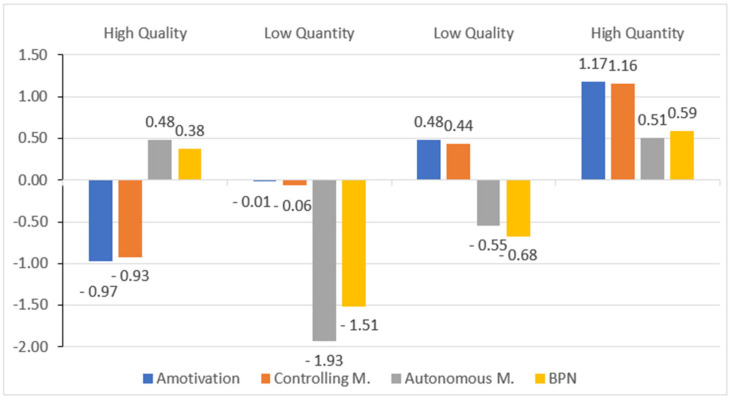
Student motivational profiles.

**Table 1 children-10-00112-t001:** Descriptive analysis, reliability and correlations.

	Variables	*M*	*SD*	A	K	α	Ω	1	2	3	4	5	6	7	8	9	10	11	12	13	14	15
1.	Autonomy support	5.79	1.08	−1.50	3.05	0.85	0.87	-														
2.	Competence	5.11	0.83	−0.87	0.40	0.89	0.89	0.41 **	-													
3.	Autonomy	4.94	1.03	−0.81	0.10	0.87	0.87	0.27 **	0.47 **	-												
4.	Relatedness	5.02	0.97	−0.79	0.01	0.86	0.87	0.34 **	0.50 **	0.70 **	-											
5.	General IM	6.12	0.96	−1.18	1.22	0.86	0.86	0.47 **	0.63 **	0.36 **	0.46 **	-										
6.	Knowledge IM	6.11	0.99	−1.21	1.16	0.90	0.90	0.48 **	0.65 **	0.34 **	0.45 **	0.90 **	-									
7.	Stimulation IM	6.01	1.00	−1.07	0.94	0.89	0.89	0.47 **	0.65 **	0.37 **	0.46 **	0.87 **	0.91 **	-								
8.	Achievement IM	6.30	0.75	−1.13	0.98	0.78	0.78	0.47 **	0.65 **	0.38 **	0.47 **	0.86 **	0.88 **	0.88 **	-							
9.	Integrated R	6.30	0.96	−1.12	0.93	0.78	0.78	0.47 **	0.65 **	0.38 **	0.47 **	0.85 **	0.88 **	0.87 **	0.99 **	-						
10.	Identified R	6.14	0.93	−1.12	0.81	0.87	0.87	0.46 **	0.63 **	0.34 **	0.44 **	0.84 **	0.88 **	0.87 **	0.86 **	0.86 **	-					
11.	Introjected R	3.65	2.14	0.02	−1.42	0.93	0.93	−0.01	−0.01	0.01	0.02	−0.04 *	−0.02	0.01	−0.03	−0.03	−0.02	-				
12.	External R	3.32	2.10	0.27	−1.32	0.93	0.93	−0.02	−0.03	0.01	0.01	−0.08 **	−0.07 **	−0.03	−0.08 **	−0.08 **	−0.08 **	0.90 **	-			
13.	Amotivation	3.34	2.13	0.20	−1.41	0.94	0.94	−0.06 **	−0.07 **	−0.03	−0.02	−0.11 **	−0.10	−0.07 **	−0.11	−0.11 **	−0.10 **	0.81 **	0.83 **	-		
14.	IPA	4.35	0.66	−1.04	0.60	0.80	0.80	0.34 **	0.52 **	0.31 **	0.37 **	0.66 **	0.67 **	0.68**	0.64 **	0.64 **	0.66 **	0.02	0.01	−0.02	-	
15.	Enjoyment	4.44	0.67	−1.43	2.16	0.87	0.87	0.36 **	0.54 **	0.29 **	0.38 **	0.71 **	0.71 **	0.71**	0.68 **	0.68 **	0.69 **	−0.05 **	−0.08 **	−0.11 **	0.74 **	-
16.	Boredom	2.88	1.09	0.32	−0.53	0.65	0.66	0.00	−0.01	0.02	0.01	−0.05 **	−0.05 **	0.00	−0.05 **	−0.05 **	−0.04 *	0.67 **	0.71 **	0.65 **	0.04 *	−0.05 *

Note: * *p* < 0.05; ** *p* < 0.01; M = Mean; SD = Standard Deviation; A = Asymmetry; K = Kurtosis; α = Cronbach Alpha Value; Ω: omega; IM = intrinsic motivation; R = Regulation; IPA = Intention to be physical active.

**Table 2 children-10-00112-t002:** Differences in the variables of the motivational profiles.

	High Quality *n* = 1094		Low Quantity *n* = 292		Low Quality *n* = 555		High Quantity *n* = 680				
	*M*	*SD*	*M*	*SD*	*M*	*SD*	*M*	*SD*	*F*	*p*	*η²*
Amotivation	−0.97	0.30	−0.01	0.68	0.47	0.48	1.17	0.48	3481.68	<0.01	0.80
Controlling Motivation	−0.93	0.50	−0.06	0.62	0.44	0.44	1.16	0.50	2559.78	<0.01	0.74
Autonomous Motivation	0.48	0.57	−1.93	0.76	−0.55	0.66	0.51	0.49	1568.42	<0.01	0.64
BPN	0.38	0.76	−1.51	0.85	−0.68	0.69	0.59	0.54	877.79	<0.01	0.50
Box’s = 1060.08 (f = 35.21) *p* < 0.01											
Pillai’s trace = 1.55 (f = 699.09) *p* < 0.01											

Note: *n* = Number of participants; *M* = Mean, *SD* = Standard Deviation; BPN = Basic psychological needs; η² = Partial eta squared.

**Table 3 children-10-00112-t003:** Differences in autonomy support, intention to be physically active, enjoyment and boredom according to the motivational profiles.

	High Quality *n* = 1094		Low Quantity *n* = 292		Low Quality *n* = 555		High Quantity *n* = 680				
	*M*	*SD*	*M*	*SD*	*M*	*SD*	*M*	*SD*	*F*	*p*	*η²*
Autonomy support	0.26	0.86	−1.06	0.96	−0.32	0.84	0.30	0.97	215.55	<0.01	0.19
IPA	0.31	0.75	−1.44	1.03	−0.42	0.87	0.46	0.70	498.48	<0.01	0.36
Enjoyment	0.40	0.63	−1.54	1.21	−0.42	0.88	0.36	0.64	590.69	<0.01	0.40
Boredom	−0.62	0.76	−0.07	0.67	0.24	0.71	0.83	0.99	466.57	<0.01	0.34
Box’s = 551.04 (f = 18.30) *p* < 0.01											
Pillai’s trace = 0.83 (f = 251.79) *p* < 0.01											

Note: *M* = Mean; *SD* = Standard Deviation; η² = Partial eta squared; IPA = intention to be physically active.

**Table 4 children-10-00112-t004:** Contrast of multiple comparisons according to Z values.

	1 vs. 2	1 vs. 3	1 vs. 4	2 vs. 3	2 vs. 4	3 vs. 4
Autonomy support	1.32 **	0.57 **	−0.04	−0.74 **	−1.36 **	−0.61 **
IPA	1.75 **	0.72 **	−0.15 **	−1.02 **	−1.90 **	−0.87 **
Enjoyment	1.94 **	0.82 **	0.03	−1.11 **	−1.90 **	−0.78 **
Boredom	−0.55 **	−0.85 **	−1.44 **	−0.30 **	−0.89 **	−0.59 **

Note: 1: high quality profile; 2: low quantity profile; 3: low quality profile; 4: high quantity profile, IPA: intention to be physically active; ** *p* < 0.01.

**Table 5 children-10-00112-t005:** Differences according to gender, age and extracurricular activity.

	High Quality			Low Quantity			Low Quality			High Quantity					
	*n*	%	R	*n*	%	R	*n*	%	R	*n*	%	R	χ^2^	gl	*p*
Men	532	48.60%	−0.9	140	47.90%	−0.6	258	46.50%	−1.7	373	54.90%	3.1	10.37	3	0.02
Women	562	51.40%	0.9	152	52.10%	0.6	297	53.50%	1.7	307	45.10%	−3.1			
8–12	302	27.60%	4.0	52	17.80%	−2.5	109	19.60%	−2.5	157	23.10%	−0.4	25.43	6	0.00
13–15	494	45.20%	−1.4	144	49.30%	0.9	255	45.90%	−0.5	334	49.10%	1.4			
16–18	298	27.20%	−2.2	96	32.90%	1.3	191	34.40%	2.8	189	27.80%	−1.2			
ESP Yes	661	60.40%	4.3	133	45.50%	−3.6	248	44.70%	−5.7	411	60.40%	3.1	55.42	3	0.00
ESP No	433	39.60%	−4.3	159	54.50%	3.6	307	55.30%	5.7	269	39.60%	−3.1			

Note: *n* = number of participants; % = participants percentage; *R* = corrected standardized residuals; χ^2^ = chi squared; ESP = extracurricular sports practice.

## Data Availability

Data collected and analyzed during the study are available upon reasonable request.

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
