# Peer review of "Motivational Profiles in Physical Education: Differences at the Psychosocial, Gender, Age and Extracurricular Sports Practice Levels"

_children, 2023, doi:10.3390/children10010112_

Round 1

Reviewer 1 Report

Very interesting research dealing with issues not so often undertaken in research programs. The relatively large size in the study group allows for the generalization of the results of the research.

However, with the age range from 8 to 18 years, the respondents' opinions should be divided into age categories, because the opinions of an 8-year-old and an 18-year-old child are completely different, for example due to developmental needs.

In Figure 1, there is no need for a table with data, and you should put figures on bars.

Author Response

Very interesting research dealing with issues not so often undertaken in research programs. The relatively large size in the study group allows for the generalization of the results of the research.

However, with the age range from 8 to 18 years, the respondents' opinions should be divided into age categories, because the opinions of an 8-year-old and an 18-year-old child are completely different, for example due to developmental needs.

REPLY: We totally agree with the suggestion the age range should be divided into age categories. Indeed, we have already done from 8 to 12, from 13 to 15 and from 16 to 18 years old. In this regard, we have divided the sample considering the gender and the extracurricular activity.  Adding three more columns to the table 1 could make it too big and not easy to handle. Furthermore, we should include four more columns for the gender and extracurricular activity.

In Figure 1, there is no need for a table with data, and you should put figures on bars.

REPLY: The table in Figure 1 has been deleted and figures have been put on bars.

Reviewer 2 Report

Thanks for opportunity review manuscript entitled ''Motivational profiles in Physical Education: differences at the  psychosocial, gender, age and extracurricular sports practice  levels.'' The article is well written. However, some  revisions required before publication of article. I can summarize necessary revisions as follows:

1. The title must be corrected. the correct form is Motivational profiles in physical education: Differences at the  psychosocial, gender, age and extracurricular sports practice  levels.

2. All small ns representing frequencies must be italic. Moreover, mean and standard deviation must be italic. 

3. Manuscript, General: The main problem in he manuscript is construction of paragraphs. Author must construct each paragraph with three to eight sentences as per APA 7 rules. 

4. Introduction, general:  Three  weakness exist in the Introduction. First is related to previous studies, second is related to Spanish cultural context. Firstly, author must give information about previous studies that examine motivational profiles in physical education students using cluster analyses and their weakness ıf available. 

5. Introduction, general: Second weakness is that authors need to answer importance of their study by answering following question in Introduction section ' 'Why is it important to examine motivational profiles in physical education and  differences at the  psychosocial, gender, age and extracurricular sports practice  levels in Spanish cultural context? Authors did not tell anything about it. 

6. Introduction, general: Literature review must be improved in Introduction section authors only summarized hem with a single paragraph. 

7. Introduction, general: Author must add research questions or their hypothesis to after aim sentence. 

8. Method, Participants section: All small ns representing frequencies must be italic. Moreover, mean and standard deviation must be italic. 

9. Method, Scales section: No need following and must remove ' 'In the present study, a questionnaire of closed questions was used consisting of two  parts, the first contained sociodemographic variables, and the second part the scales used  in the study, described below:

10. Method, Scales section: There is a mistake in Autonomy support section on Line 118-119 and must be corrected. Following ‘ ‘. completely agree)

11. Method, Scales section: Author must add possible scores, meaning of higher scores and response options (e.g., completely disagree (1) to completely agree (5)) of The Learning Climate Questionnaire

12. Method, Scales section: (LCQ; [34]  must be corrected as (LCQ; [34])

13. Method, Scales section: Following is unclear The internal consistency revealed a value of α = .85, Ω = .87 in this study or original study? . This comment is valid for all scales in method section. Authors must clarify this problem.

14. Method, Scales section: Authors must add possible scores, meaning of higher scores of The Psychological Need Satisfaction in Exercise Scale (PNSE).

15. Method, Scales section:  Authors must add possible scores, meaning of higher scores of Sport Conduct Regulation Questionnaire (BRSQ)

16. Method, Scales section:  Authors must add possible scores, meaning of higher scores of Intention to be physically active

17. Method, Scales section:  Authors must add possible scores, meaning of higher scores of Sport Satisfaction Instrument.

18. Method, Scales section:  Demographic information form is completely missing in scales section and must be added.

19. Method, Data analysis section:  Following is wrong and must be corrected ‘ ‘In addition, the effect size was calculated in terms of partial eta -squared (η2), considering a small effect size with values <.01, a medium effect between .01 217 and .06, and a large effect with a value >.14 ’’ The true is  .01-.05 = small, .06-.13, medium, .14 or above large.

20. Method, Data analysis section:  Following sentence is wrong and must be corrected ‘ ‘establishing a level of significance p < .05.’’

21. Manuscript, Results General: Authors sometimes reported two, sometimes reported findings with three decimal. Authors must consistently choose one and report them. I recommend two decimal as per APA 7 rules.

22. Results, Table 1: Legend must be Note. Following is wrong and must be corrected ‘ ‘A = Asimmetru; K = lirtpsos’’ Repor small to lrge not like these, **p < .01; *p < .05 correct is *p < .05; **p < .01.

23. Results, correlation section: authors must report correlations as significantly positively, or significantly negatively, authors did not tell anything about direction or magnitude. (moderately positively correlated with…..)

24. Results, Table 2-Table 3: Authors must add sybol of partial eta-square to Tables (writing as eta). In this form it seems as if eta-square and they are completely different.

25. Result, Line 269-271: It is very difficult to understand following, Please rewrite ‘ ‘Table 2 shows the differences between the variables that were part of the cluster 268 solution. They had a multivariate effect (Box value = 1060.080, f = 35.213, p < .001), sug- 269 gesting the use of the Pillai trace as a test statistic [50], showing a value of 1.550 (f = 270 699.092) p < .001.’’

26. Results, Table 1-Table 5: All small ns representing frequencies must be italic. Moreover, mean and standard deviation must be italic in the tables.

27. Results, Line 301-303: Author must check correctness of following ‘ ‘Using the corrected standardized residuals gives us information on where these differences are found, since residuals equal to  or greater than 1.90 are considered as indicators that there are dependent of these two  categories and, therefore, the differences are significant’’ I think it must be 1.96.

28. Discussion, General: Practical implications of study findings is completely missing and must be added.

29. Discussion, General: Authors did not interpret any findings and must interpret some of them at least.

30. Results, general: homogeneity of variance assumption clearly violated in these analyses. Aıuthors must clarify what they do to handle this problem.

31. Apart from above corrections, the article is original and generally well-written. I congratulate them.  

Author Response

Thanks for opportunity review manuscript entitled ''Motivational profiles in Physical Education: differences at the  psychosocial, gender, age and extracurricular sports practice  levels.'' The article is well written. However, some  revisions required before publication of article. I can summarize necessary revisions as follows:

  1. The title must be corrected. the correct form is Motivational profiles in physical education: Differences at the  psychosocial, gender, age and extracurricular sports practice  levels.

REPLY: It has been changed.(line 2).

  1. All small ns representing frequencies must be italic. Moreover, mean and standard deviation must be italic. 

REPLY: It has been corrected throughout the document.

  1. Manuscript, General: The main problem in the manuscript is construction of paragraphs. Author must construct each paragraph with three to eight sentences as per APA 7 rules. 

REPLY: The document has been adjusted between 3 to 8 sentences according to the given recommendations.

  1. Introduction, general:  Three weakness exist in the Introduction. First is related to previous studies, second is related to Spanish cultural context. Firstly, author must give information about previous studies that examine motivational profiles in physical education students using cluster analyses and their weakness ıf available. 

REPLY: It is specified through the studies 7, 8 and 9 with cluster analysis (lines 45-46).

Additionally, 9, 19, 30, 32 studies are placed between lines 78-85.

In addition, 17, 23,24 studies detailed in the introduction cover motivational profiles in PE students through the cluster analysis.

  1. Introduction, general: Second weakness is that authors need to answer importance of their study by answering following question in Introduction section ' 'Why is it important to examine motivational profiles in physical education and  differences at the  psychosocial, gender, age and extracurricular sports practice  levels in Spanish cultural context? Authors did not tell anything about it. 

REPLY: The revisor’s suggestions have been included in a paragraph in lines 87-91.

  1. Introduction, general: Literature review must be improved in Introduction section authors only summarized hem with a single paragraph. 

REPLY: Information has been added in paragraphs placed between lines 75-86 and 87-91.

  1. Introduction, general: Author must add research questions or their hypothesis to after aim sentence. 

REPLY: The hypothesis has been indicated after the objective in lines 99-105.

  1. Method, Participants section: All small ns representing frequencies must be italic. Moreover, mean and standard deviation must be italic. 

REPLY: It has been changed. (Lines 117-119, 122-124).

  1. Method, Scales section: No need following and must remove ' 'In the present study, a questionnaire of closed questions was used consisting of two  parts, the first contained sociodemographic variables, and the second part the scales used  in the study, described below:

REPLY: It has been removed.

  1. Method, Scales section: There is a mistake in Autonomy support section on Line 118-119 and must be corrected. Following ‘ ‘. completely agree)

REPLY: It has been corrected (line 132).

  1. Method, Scales section: Author must add possible scores, meaning of higher scores and response options (e.g., completely disagree (1) to completely agree (5)) of The Learning Climate Questionnaire

REPLY: It has been fixed as suggested. (Line 133).

  1. Method, Scales section: (LCQ; [34]  must be corrected as (LCQ; [34])

REPLY: It has been changed. (Line 128).

  1. Method, Scales section: Following is unclear The internal consistency revealed a value of α = .85, Ω = .87 in this study or original study? . This comment is valid for all scales in method section. Authors must clarify this problem. 

REPLY: Values belonging to the present study have been clarified in all the section scales of the method (lines 133,143-44,166,179,191).

  1. Method, Scales section: Authors must add possible scores, meaning of higher scores of The Psychological Need Satisfaction in Exercise Scale (PNSE).

REPLY: This information is shown in line 142.

  1. Method, Scales section:  Authors must add possible scores, meaning of higher scores of Sport Conduct Regulation Questionnaire (BRSQ)

REPLY: This information is shown in lines 161-162.

  1. Method, Scales section:  Authors must add possible scores, meaning of higher scores of Intention to be physically active

REPLY: This information is shown in lines 177-178.

  1. Method, Scales section:  Authors must add possible scores, meaning of higher scores of Sport Satisfaction Instrument.

REPLY: This information is shown in lines 189.

  1. Method, Scales section:  Demographic information form is completely missing in scales section and must be added. 

REPLY: Information has been added in the last paragraph of the scale section (line 192-195).

  1. Method, Data analysis section:  Following is wrong and must be corrected ‘ ‘In addition, the effect size was calculated in terms of partial eta -squared (η2), considering a small effect size with values <.01, a medium effect between .01 217 and .06, and a large effect with a value >.14 ’’ The true is  .01-.05 = small, .06-.13, medium, .14 or above large. 

REPLY: It has been corrected as suggested. (line 237-238).

  1. Method, Data analysis section:  Following sentence is wrong and must be corrected ‘ ‘establishing a level of significance p < .05.’’

REPLY: It has been corrected (line 243).

  1. Manuscript, Results General: Authors sometimes reported two, sometimes reported findings with three decimal. Authors must consistently choose one and report them. I recommend two decimal as per APA 7 rules. 

REPLY: It has been corrected throughout the document according to the suggestion to choose two decimals. 

  1. Results, Table 1: Legend must be Note.Following is wrong and must be corrected ‘ ‘A = Asimmetru; K = lirtpsos’’ Repor small to lrge not like these, **p < .01; *p < .05 correct is *p < .05; **p < .01. 

REPLY: It has been fixed as suggested (Table 1).

  1. Results, correlation section: authors must report correlations as significantly positively, or significantly negatively, authors did not tell anything about direction or magnitude. (moderately positively correlated with…..)

REPLY: It has been corrected. (Line 251,255).

  1. Results, Table 2-Table 3: Authors must add sybol of partial eta-square to Tables (writing as eta). In this form it seems as if eta-square and they are completely different. 

REPLY: It has been fixed as suggested (Table 2-Table 3).

  1. Result, Line 269-271: It is very difficult to understand following, Please rewrite ‘ ‘Table 2 shows the differences between the variables that were part of the cluster 268 solution. They had a multivariate effect (Box value = 1060.080, f = 35.213, p < .001), sug- 269 gesting the use of the Pillai trace as a test statistic [50], showing a value of 1.550 (f = 270 699.092) p < .001.’’

REPLY: It has been corrected. (Line 289-292).

  1. Results, Table 1-Table 5: All small ns representing frequencies must be italic. Moreover, mean and standard deviation must be italic in the tables. 

REPLY: It has been fixed as you suggested in all the tables.

  1. Results, Line 301-303: Author must check correctness of following ‘ ‘Using the corrected standardized residuals gives us information on where these differences are found, since residuals equal to  or greater than 1.90 are considered as indicators that there are dependent of these two  categories and, therefore, the differences are significant’’ I think it must be 1.96. 

REPLY: The value has been corrected (Line 329), appreciate the clarification.

  1. Discussion, General: Practical implications of study findings is completely missing and must be added. 

REPLY: A paragraph between lines 402-409 and 410-418 has been added same that specified the practical applications.

  1. Discussion, General: Authors did not interpret any findings and must interpret some of them at least. 

REPLY: A paragraph between lines 402-409 and 410-418 has been added same that specified the importance of the findings so it can have a practical application.

  1. Results, general: homogeneity of variance assumption clearly violated in these analyses. Aıuthors must clarify what they do to handle this problem. 

REPLY: The Box M and Levene test to measure the homogeneity of variance in the de-pendent variable in all the groups defined by the factors, were carried out observing that only in the case of the autonomy support variable the p-values were greater than .05, so there was no homogeneity of variances for the rest of the variables, specifically the mo-tivation variables (controlling and autonomous motivation, and amotivation), the basic psychological needs, enjoyment and boredom, and intention to be physically active. This leads us to indicate not all the initial hypotheses were satisfied, and therefore, that the results are not completely conclusive.  This information has been included in the manuscript, specifically in the section 3.3. Analysis of differences among cluster, lines 295-302.

  1. Apart from above corrections, the article is original and generally well-written. I congratulate them.  

 REPLY: We appreciate the opportunity to correct this article.

Reviewer 3 Report

Generally speaking, this is an interesting and imperative topic about students motivational in physical education, and even though, some valuable findings can be concluded from the present work, it is a pity that two serious errors has been also generated. On the one hand, age of the participants in this work is from 8 to 18 years old, which contains the elementary schools, the junior high schools, and the senior high schools, so the range of these participants are so wide. Due to the obvious different motivational profiles for the three groups, and at the same time, these subjects are not distinguished by the authors, so that the research results reliability are relatively low. On the other hand, it can be found that the subjects in this work are from Ecuador, but all instruments are used by the Spanish version. I wonder why so evident cultural context difference between two countries hasnt been considered in this work?Hence, the validity of all instruments are doubtful.

Except for those, several faults are also needed to be mentioned. Firstly, in table 1, SD for variable 11 to 13, and 16, are too large, which means the great variability of those data. Secondly, all values in this work should retain two decimals, but they are not united now. Thirdly, in Line 196-198,that expression about normality of data is informal, and the values of asymmetry and kurtosis were analyzed should be revised as follows: the values of skewness and kurtosis were analyzed, as well as the sign eTa should be corrected as partial η2. Fourthly, the omega value for all instruments are excessive and unnecessary, and the letter legend below all tables maybe an mistake, and it can be changed to Note. In addition, table 2 and table 4 are cross-page, and the sign of chi-square in table 5 is wrong, which is a special sign χ rather than a capital X. No doubt, there are so many errors are existed in the present research. A t last, the conclusion is redundant and tedious, the author should tell us the main findings instead of some detailed content. 

Author Response

Generally speaking, this is an interesting and imperative topic about students’ motivational in physical education, and even though, some valuable findings can be concluded from the present work, it is a pity that two serious errors has been also generated.

On the one hand, age of the participants in this work is from 8 to 18 years old, which contains the elementary schools, the junior high schools, and the senior high schools, so the range of these participants are so wide. Due to the obvious different motivational profiles for the three groups, and at the same time, these subjects are not distinguished by the authors, so that the research results’ reliability are relatively low.

REPLY: About age it was distinguished according to the scholar Ecuadorian system into three groups: 8 – 12 years old, 13 – 15 years old, and 16 – 18 years old, basic school, elemental school, high school respectively.

On the other hand, it can be found that the subjects in this work are from Ecuador, but all instruments are used by the Spanish version. I wonder why so evident cultural context difference between two countries hasn’t been considered in this work?Hence, the validity of all instruments are doubtful.

REPLY: The instruments validated in Spanish were used and to prevent the cultural context from leading the respondents to misinterpret some of the items that were part of the questionnaire, the researcher was always explaining each of the scales prior to completion, solving doubts of those items that could generate some confusion.

Except for those, several faults are also needed to be mentioned. Firstly, in table 1, SD for variable 11 to 13, and 16, are too large, which means the great variability of those data. Secondly, all values in this work should retain two decimals, but they are not united now. Thirdly, in Line 196-198,that expression about normality of data is informal, and “the values of asymmetry and kurtosis were analyzed” should be revised as follows: “the values of skewness and kurtosis were analyzed”, as well as the sign “eTa” should be corrected as “partial η2”. Fourthly, the omega value for all instruments are excessive and unnecessary, and the letter “legend” below all tables maybe an mistake, and it can be changed to “Note”. In addition, table 2 and table 4 are cross-page, and the sign of chi-square in table 5 is wrong, which is a special sign “χ” rather than a capital “X”. No doubt, there are so many errors are existed in the present research.

REPLY: Regard to SD these are the results of this study and indeed are bigger that the mention variables.

Values have been adjusted into two decimals along the document.

In line 218 have been corrected as suggested “the values of skewness and kurtosis were analyzed”.

The sign η2 has been corrected (Table 2 and Table 3).

The omega value has been used in many studies like this one, none of the other to revisors have 

mentioned this. Nevertheless, if you consider it is needed it could be eliminated in all the document.

The word “legend” below all tables has been changed by “note”.

Tables 2 and 4 table are cross-page have been adjusted.

The chi square sign has been corrected (table 5).

At last, the conclusion is redundant and tedious, the author should tell us the main findings instead of some detailed content.

REPLY: The conclusion has been adjusted making emphasis in the main findings (Lines 431-442).

Round 2

Reviewer 2 Report

Authors revised the article with a good will. I am satisfied with the revisions. I recommend accept decision. However, I recommend authors that during production process, please add a statement that indicating official language of Ecuador is Spanish. thus you used Spanish version of scales preferably above 2.3.1 Autonomy support below 2.3 Instruments. 

Reviewer 3 Report

The most key point about age range and measurement tools have not been resolved by authors.